# The Effectiveness of Negative Pressure Therapy: Nursing Approach

**DOI:** 10.3390/jpm12111813

**Published:** 2022-11-01

**Authors:** Álvaro Astasio-Picado, María Dolores Murcia Montero, Miriam López-Sánchez, Jesús Jurado-Palomo, Paula Cobos-Moreno, Beatriz Gómez-Martín

**Affiliations:** 1Physiotherapy, Nursing and Physiology Department, Faculty of Health Sciences, University of Castilla-La Mancha, 45600 Toledo, Spain; 2Extremadura Health Service, 10300 Cáceres, Spain; 3Nursing Department, University of Extremadura, 10600 Plasencia, Spain

**Keywords:** negative pressure therapy, surgical wound, surgical wound infection, nursing, pressure ulcers, diabetic foot, dehiscence

## Abstract

Introduction: Complex wounds require advanced techniques for their management and care. Wound care costs are high, so healthcare professionals need to be aware of available therapies. Negative pressure therapy is a technology for which more and more data on its effectiveness in complex wounds are being collected. Objective: The objectives of this review were to analyze if the application of negative pressure therapy in complex wounds is effective; to compare the effectiveness of negative pressure therapy with other conventional treatments, as well as its combination with other therapies; and to evaluate the quality of life of patients undergoing negative pressure therapy and collect their main characteristics. Methodology: A bibliographic review focused on articles published between November 2015 and June 2022 was carried out. The following databases were consulted: PubMed (Medline), Google Scholar, Web of Science (WOS), Scielo and Scopus. Results: The most used pressures in the studies coincide at −125 mmHg and in the range of −125 mmHg to −150 mmHg. In the pediatric population, pressure levels vary by age group. A pressure of −75 to −125 mmHg is recommended for children over 12 years of age, and −50 to −75 mmHg is recommended for children under 2 years of age. Conclusions: Negative pressure therapy stands out for its rapid rate of granulation, the prevention and effective treatment of infections, the variety and malleability of dressings, its various applications and the possibility of using it with other therapies to accelerate wound closure.

## 1. Introduction

Chronic wounds are a public health problem that compromises many aspects. The care of chronic wounds is a challenge for nurses, the professionals in charge of its approach, since it requires a large part of their work time [1,2]. Research and knowledge about the biology of wounds; healing processes; and the physiology of the microenvironment of the lesion from the molecular, infectious and genetic points of view, as well as the appearance of technologies, allow us to critically and precisely address its approach [3].

Given the characteristics of the patients, a large percentage of these wounds tend not to heal spontaneously [3]. This leads to their chronicity, as they become highly complex and long-lasting wounds whose treatment requires advanced care techniques [4]. These injuries produce great stress in all people involved in their course. They are a considerable cause of morbidity, increase disability and reduce the quality of life of those involved by 0.5 QALYs per year [5].

The financing of this care is high, even multiplying; therefore, it is important that the professionals responsible for the use of resources are aware of the availability of viable therapies. This will help them choose the one that best suits each case in order to reduce healing time, improve the patient’s quality of life and limit healthcare costs [2,4].

One of the treatments on the rise for complex wounds is negative pressure therapy (NPT). This treatment is an active system that is being driven by its good results, although it is an apparently expensive technology whose usefulness and location are still largely unknown [4].

### 1.1. Negative Pressure Therapy Concept

Negative pressure therapy (NPT), also known as vacuum-assisted closure therapy (VAC), is a system that helps promote wound healing, non-invasively, by applying localized negative pressure to the wound. It is used both in hospital and outpatient settings [2]. Its airtight system provides a moist, closed environment that removes excess fluids, promoting wound healing. Its objective is based on the epithelialization of the wound, which has not been achieved by the first intention, promoting the evolution of the lesion from the inflammatory to the proliferative phase [2].

### 1.2. Pathophysiological Effects of Negative Pressure Therapy

The foam dressings that are in contact with the wound have a regular structure with open pores of considerable size (400–600 µm) allowing the pressure to be the same throughout the wound. The pressure causes the volume of the foam to reduce, leading to cell expansion, wound contraction and fluid removal [6].

Its mechanism of action allows for reducing the excess of liquid and with it the edema, the exudate, the inflammation and the cellular detritus. This reduces the bacterial load and promotes blood and lymphatic flow in a localized way. In addition, it allows neovascularization, increasing oxygenation and the supply of nutrients that cells need [2].

The device is capable of modifying the cytoskeleton of cells, based on intercellular signals that increase cell division, which allows borders to approach. This prepares the wound bed by forming granulation tissue and promoting epithelial growth factors [2].

### 1.3. Indications and Contraindications of Negative Pressure Therapy

In general, the manufacturer recommends the use of negative pressure therapy on wounds that have not decreased by approximately 50% in size within a month with conventional treatment. Its applications are multiple, although for this it is essential that the patient is hemodynamically stable [6,7].

The scientific evidence for its application [6,7,8] and contraindication [2,6,8] is found in the cases shown in Table 1.

### 1.4. Complications of Negative Pressure Therapy

Complications can be avoided with good technique, management and vigilance. Among the most common complications we find bleeding during sponge change due to excessive growth of granulation tissue, as well as problems with odor. Necrosis of the tissue appears on the skin at the edges of the wound. Similarly, infection, toxic shock syndrome and fluid depletion appear. Pain is common during dressing changes and when negative pressure is restored after the change [8,9].

### 1.5. Management of Negative Pressure Therapy

After confirming the viability of the application of NPT and the patient’s acceptance of the informed consent, the person in charge will follow the following steps [3,8,10]:The first step is to comply with the appropriate asepsis measures.Start by cleaning the wound with 0.9% FSS together with a gauze or compress using a drag strategy. If the wound presents granulation tissue with abundant bleeding, a hydrocolloid or silicone mesh dressing will be placed on that area to prevent it from sticking to the sponge. If the wound has necrotic tissue, it should be debrided beforehand.The surface of the wound will be measured to cut and mold the sponge according to its extension, caliber, type and volume of exudate.The perforated silicone probe will be placed inside the sponge without touching the tissue to avoid bleeding.The perilesional skin, after being cleaned and dried, will be protected with a silicone–acrylic copolymer barrier or with hydrocolloid dressings.The skin along the path where the probe will be fixed will also be protected. In this way, we prevent friction injuries or pressure ulcers.Once the fenestrated sponge with the evacuation probe has been placed, it will be connected, on the one hand, with the container that will collect the waste and, on the other hand, with the controlled suction pump to start.The area will be covered by a transparent adhesive dressing that will cover up to 3–5 cm of the tissue surrounding the injury.

The therapy will be maintained for a maximum of 22 h a day, as it requires a rest period to cause a local decrease in interstitial pressure [8,9]. This disconnection can be used during the patient’s activity, such as mealtime or going for a walk, so that the patient can have a certain degree of autonomy [10,11,12,13].

The objective of this study is to analyze the effectiveness of NPT in its various applications in complex wounds.

## 2. Materials and Methods

The preparation of this work was carried out through a systematic bibliographic review of the articles found by searching the following databases: Medline/PubMed, WOS, Scielo, Scopus and Google Scholar. To find the best possible scientific evidence, a series of inclusion and exclusion criteria were applied.

### 2.1. Information Sources and Search Strategy

The keywords for this review are: ventilator-associated pneumonia; nosocomial; nursing; prevention; preventive measures; microorganisms; knowledge. To carry out the bibliographic search, different keywords in English were used, such as: “Negative-Pressure Wound Therapy”, “Surgical Wound”, “Nursing Care”, “Surgical Wound Dehiscence”. These have been validated by the DeCS and MeSH. Once the keywords were selected, the corresponding Boolean operators AND/OR were used, as were the necessary parentheses and quotation marks. The final search string is as follows: (Negative-Pressure Wound Therapy OR Negative-Pressure Wound Therapies OR Vacuum Assisted Closure) AND (Wound infection OR Infection, Wound) AND (Wound Healing OR Healing, Wound) AND (Nursing Care OR Care, Nursing OR Management, Nursing Care).

### 2.2. Inclusion Criteria and Exclusion Criteria

The criteria that were taken into account for the selection of the relevant studies were the following: Inclusion criteria: the period between 2015 and 2022; article type: article review and article research; field: medicine; English language; sample in adult population; studies that provide scientific evidence justified by the level of indexing of articles in journals according to the latest certainties. Exclusion criteria: articles prior to 2015; language: not English; studies in which the population was minors; studies that do not provide scientific evidence justified by the level of indexing of articles in journals according to the latest certainties.

### 2.3. Methodological Evaluation of the Data Used

For the methodological evaluation of the individual studies and the detection of possible biases, the evaluation was carried out using the PEDro Evaluation Scale. This scale consists of 11 items, providing one point for each element that is fulfilled. The articles that obtained a score of 9–10 points have an excellent quality, those that obtained between 6 and 8 points have a good quality, those that obtained 4–5 points have an intermediate quality and those that obtained less than 4 points have a poor methodological quality (Appendix A) [14,15].

The Scottish Intercollegiate Guidelines Network classification was used in the data analysis and assessment of the levels of evidence, which focused on the quantitative analysis of systematic reviews and the reduction of systematic error. Although it took into account the quality of the methodology, it did not assess the scientific or technological reality of the recommendations (Appendix B) [16].

## 3. Results

The research question was constructed following the PICO format (population/patient, intervention, comparator and outcomes/outcomes), detailed as P (patients): subjects of any sex and age with complex wounds; I (intervention): negative pressure therapy (NPT) or vacuum-assisted closure (VAC) system; C (comparison): conventional wound healing techniques; O (results): effectiveness, comparison of results with other treatments and combined with other treatments, quality of life, costs, complications and characteristics of pressure and dressings (Figure 1).

Below is a table that shows the search strategy used to select the 30 articles selected from the five databases, following the criteria of identified studies, duplicate studies, title, abstract, full text and valid studies of a definitive nature (Table 2). The total number of valid articles is summarized in Appendix C.

In this narrative review, the most relevant data from the 30 selected articles on treatment with NPT in complex wounds have been included.

Data on abdominal necrotizing fasciitis (28 and 55 days of NPT), Fournier’s gangrene (19, 45 and 29 days of NPT) [17], abscess, osteomyelitis, mediastinitis (11, 20, 14 and 28 days of NPT) [18,19], associations with pacemakers and cardioverter defibrillator (6 and 9 days of NPT) [20], eviscerations and perforations (6, 14, 14 and 26 days of NPT) [21] were found.

In the group of dehiscences, we can find cases of laparotomy (11 and 30 days of NPT) [19,22], sternotomy (17 days of NPT) [1] and surgical wounds (14, 20, 29, 12 and 12 days of NPT) [19,21,22,23].

The data on ulcers are differentiated into venous (8 and 8 days of NPT) [24], diabetic foot (10 days of NPT) [21] and pressure ulcers (12 days of NPT) [25], and we also found a case of necrosis (17 days) [26].

Among the burns, the NPT application time varies between 10, 13 and 14 days [18,24,27].

Exposed limb cases represent cases of NPT-treated lower extremity wounds with 9, 28, 35, 7, 12, 21, 21 and 4 days of NPT [18,22,28,29].

Regarding the different pressures reported in the cases of the articles, −125 mmHg is the most frequently used pressure, being used in 14 cases [17,18,22,26,29]. It is also worth noting the use of the −125 mmHg to −150 mmHg interval of pressure, which was repeated in six cases [22,24]. The least used pressures are −200 mmHg and −30 mmHg [21,22] (Figure 1).

Regarding the time between cures for the application of NPT, the most frequent values recorded in cases are 2 days [17,18,21,22,24] and 3 days [18,19,22,26,29]. To a lesser extent, the literature shows a spacing between cures of intervals of 2–3 days [17,18,28,30] and equal to or greater than 4 days [18,22,27,29,31] (Figure 2).

## 4. Discussion

### 4.1. Effectiveness of NPT in Complex Wounds

NPT is considered a new and effective therapy that allows faster healing and granulation results in both the adult and pediatric populations [20,24,32,33,34,35]. In a randomized study, 77.8% achieved 100% granulation at week 5 versus the control group where only 40% had reached 100% granulation [34].

The use of NPT guarantees in most of its cases an effective and faster healing; with this, we shorten hospitalization time and expenses, reducing the number of cases of complex reinterventions and use of material and human resources and thus demonstrating that it does not increase health spending [18,23,26,31,33,36]. Chu H. et al. report how the use of NPT allowed 90% healing in 10 days and complete revitalization in 17 days without complications or infections, resulting in a normotrophic scar [27].

The average time of NPT is shorter compared to conventional treatment thanks to its effectiveness, thus reducing the average time of hospitalization [17,19,20,22,23,24,27,28,29,31,37]. However, Márquez-Esppriella C. et al. did not confirm the decrease in hospital stay or the number of cures in the studied cases of the use of NPT in Fournier’s gangrene, unlike in cases of diabetic foot and transmetatarsal post-amputees [32]. On the other hand, in some studies, the wound began to be treated with conventional techniques and then NPT was chosen alone or combined with better results [18,21,26].

Regarding the costs of applying NPT, Dowsett C. et al. calculated a mean total cost per case of GBP 818 and a mean cost of GBP 24.33 in materials and GBP 13.83 in nursing per day [30]. However, in their study, González-Rubio S. et al. tested a cheaper NPT system with which an efficacy similar to that of patented NPT systems was achieved. However, the cost reduction was very significant (MXN 589 with the new design vs. MXN 11,800 with the V.A.C.) [24].

In studies such as the one by Yu L. et al., NPT is used as an infection prevention treatment, reducing surgical site infection and wound complications by 55% [38]. Various authors verify the effectiveness of NPT on infected wounds [20,22,24,33,35,38]. In addition, in the treatment of device-associated infections, there have been no cases of reinfection after withdrawal of NPT following the incorporation of a protocol: complete removal, cleaning, anterior and posterior capsulotomy and the application of NPT [20]. In addition, Garrigós X. et al. showed how the combination of NPT with intermittent instillation of antiseptics is effective for infected and contaminated wounds, reducing biofilm in open wounds [29].

In this line, an observational study recorded a mortality of 3.2% due to deep infections treated with NPT compared to conventional treatment, for which a mortality of 34.8% was recorded [39]. Other studies state that no deaths associated with NPT are reflected [20,21,23].

Lastly, Lone AM. et al. reported a notable reduction in exudate in 44.4% of cases with NPT at week 4, without this occurring in the group receiving conventional treatment [34]. In addition, Schmitz M. and Limongelli P. et al. found that NPT led to a reduction in pain as well as a decrease in redness and maceration [40,41]. However, the latter authors compared the results of conventional treatment and NPT and did not record differences in the rate of bleeding and infection [41].

### 4.2. The Effectiveness of NPT in Combination and Comparison with Other Conventional Treatments in Complex Wounds

Most authors confirm that the cost of NPT matches or is lower compared to conventional treatment [26,30,35,36]. However, there is a study that verifies that NPT has a higher cost than standard therapies when the objective is complete healing compared to when complete healing is not the goal [35].

Singh P. et al. found a faster closure rate in those patients undergoing NPT with pressures of −50 mmHg to −125 mmHg intermittently three times a day compared to patients treated with saline gauze dressings with a decrease in size, depth and volume, as well as the favorable rapid disappearance of exudate in the application of NPT. In those patients who did not present granulation tissue, it appeared at the end of the second week in 75% of the cases in which NPT was applied compared to 30% of those treated with moist gauze [36].

Likewise, Limongelli P. et al. found that patients treated with continuous suction NPT −75 mmHg subsequently replaced by incisional NPT had faster and normotrophic healing when compared to the use of hydrogen peroxide, 0.5% povidone-iodine and 1 L of saline solution, resulting in better physical and social health [41]. In addition, the studies by González-Ruiz M. and Monteiro E. et al. agree that NPT generates a larger granulation surface and faster healing than those cured with moist gauzes, and NPT healing was even achieved in half the time necessary for healing with moist gauzes [32,42].

In a systematic review on diabetic foot ulcers, it is stated that NPT helps initiate granulation tissue by reducing its size and depth and that compared to other treatments, it allows healing in less time and reduces the use of resources. However, two articles state that the efficacy of NPT is the same as that of other treatments, although it may be more effective in infected ulcers [35].

Several authors have verified the effectiveness of NPT on split-thickness skin grafts, as well as on Integra, in both pediatric and adult populations [22,24,29,34]. This combination allows close contact between the bed and the grafts, favoring their integration, reducing movement friction and facilitating drainage [29]. Likewise, Attia A. et al. show the efficacy, safety and feasibility of using NPT with the Integra Dermal Regeneration System (IDRT) followed by split-thickness skin grafting (STSG) in lower extremities, which obtained better functional and aesthetic results than those obtained with exclusive skin grafts, thus avoiding complex surgical procedures and creating another cost-effective treatment alternative [43]. In addition, Larsson JC. et al. tested the efficacy of the VAC system and dermal matrix Integra in the perineal and scrotal area and found early integration compared to those cases in which NPT was not applied [17]. Finally, Monteiro E. et al. state that NPT at −125 mmHg in continuous mode with a Granufoam dressing achieved granulation tissue 72 h after application, allowing a successful skin graft after 17 days [26].

As recorded in the article by Hajmohammadi K. et al., a necrotic and infected scalp wound previously treated with conventional treatment ended up healing completely and without complications thanks to the application of worm debridement therapy (3 weeks) with NPT and an amniotic membrane graft [14]. Likewise, García A. et al. combined NPT with dialkyl-carbamoyl chloride dressings and collagen, which reduced 50% of abdominal dehiscence in 15 days. As complications, hypergranulation was recorded with the application of collagen, and a bad smell was recorded [37].

### 4.3. Quality of Life of Patients with Complex Wounds Who Undergo NPT

Some authors speak of a higher level of satisfaction in those patients treated with NPT compared to conventional treatment [34,35,40]. The rapid recovery factor, as well as its ambulatory capacity and easy management, play an important role in improving quality of life, making it precociously functional [27,29,30,37]. In addition, Ousey KJ. et al. recognize that quality of life is more affected by wound healing than by the type of wound, so the application of NPT should be evaluated by comparing healing times [44,45].

### 4.4. Characteristics of the Use of the NPT

The most used pressures in the studies coincide at −125 mmHg and in the interval from −125 mmHg to −150 mmHg [17,18,22,24,29]. In the pediatric population, pressure levels vary between age groups, as the experts recommend using a pressure of −75 to −125 mmHg in patients over 12 years of age and −50 to −75 mmHg in patients under 2 years of age. For children between 2 and 12 years of age, it was recommended to modify the pressure depending on the location of the lesion, respecting the previous ranges. However, for sternal wounds, it is advisable to use a continuous pressure of −50 and −75 mmHg at all ages, with an exhaustive control of constants [22,23].

Dressing changes vary depending on the state and type of the wound that is being treated at any time; most frequent changes collected are recorded between 2 and 3 days [17,18,21,22,24,25,26,28,30]. Figueroa-Gutiérrez LM. et al. agree with the above but state that no more than 96 h should elapse before checking the wound and acting according to its needs [23]. On the other hand, Borrero-Esteban MP. et al. tested bacterial fluorescence to satisfactorily verify the effectiveness of NPT and establish a personalized treatment, allowing dressing changes to be guided when they were necessary, avoiding unnecessarily altering the bed [25]. In short, reducing dressing changes in burns reduces the risk of external pathogens and stops the wound healing [30].

Restrepo J. et al. and Garrigós X. et al. present several cases of abdominal wounds in which they verify that NPT is the ideal treatment for areas with poor coverage, allowing intimate dermal contact, isolating the area from the outside and facilitating the integration of mesh for the reconstruction of the area [29,31]. Among the recommended sponges is the non-adherent V.A.C. WhiteFoam dressing; this is also recommended on split-thickness skin grafting as well as on Integra without the need for intermediate dressings, registering an 80–100% adhesion after the first cure (6–7 days after its application). In cases of risk of graft loss due to superinfection such as an ostomy, it is recommended to associate it with V.A.C. GranuFoam Silver [29].

To the above, Hortelano A. et al. and Restrepo J. et al. add that if there is contact with the intestines, paraffin gauzes and WhiteFoam polyvinyl foam allow better adhesion to the tissues with very favorable results, although these require higher pressures (minimum −125 mmHg) for an adequate distribution, allowing the frequency of cures to be reduced (4–5 days) [18,31]. In abdominal dehiscence and loss of muscular support, it is advisable to use the Abdominal Dressing System [22].

Finally, to improve repositioning and extraction and avoid perilesional irritation, Fernández LG. et al. verified the use of a hybrid polyurethane dressing with acrylic adhesive and a perforated silicone layer over foam dressings [28].

In general, there are no serious complications derived from treatment with NPT [21,23]. In the articles consulted, tissue necrosis, pain and local infection are listed as frequent complications [22]. System dysfunction due to leaks is also listed as a complication in an insignificant percentage [2,3].

In an experimental study of flap implantation without subsequent application of NPT, complications included hematoma and residual seroma (10.5%), dehiscence (16%), partial loss of the flap (21%) and complete loss and epidermolysis (5% in both cases). However, the group that received NPT after flap grafting required a shorter hospitalization time due to the rapid and effective healing of all lesions [44,45,46]. In addition, two articles acknowledge that the use of NPT has been shown to reduce the number of amputations, reducing major amputations to a greater extent and not showing statistical differences in reducing minor amputations [34,35].

## 5. Conclusions

Regarding the efficacy of NPT in complex wounds, NPT systems are considered an expensive treatment compared to other conventionally used measures, so their application is very limited. However, this review has shown how the application of NPT in complex wounds has reduced hospital stays and the costs that they entail, while also reducing the consumption of material and professional resources. In addition, its application in complex wounds has helped us achieve results that could not be achieved with traditional treatment. Correctly assessing a wound to identify the most appropriate treatment is crucial for reducing costs and improving the quality of life of the patient, favoring rapid and effective recovery. Based on the efficacy of NPT in combination and comparison with other conventional treatments in complex wounds, negative pressure therapy has several properties favoring its application in wounds for which closure may be limited by the characteristics of the injury and sometimes may not be achieved with conventional treatment. Its use is extended as an effective therapy for various injuries, both those that have been complicated by a previous ineffective treatment and those for which NPT is established as the main therapy or complementary to other treatments. In patients with complex wounds undergoing NPT, quality of life has been favorably affected by its mechanism of action that allows pain reduction, collection of secretions and rapid healing, as well as its ambulatory capacity allowing the patient to be in their environment and not in a hospital, with what this entails. Regarding the characteristics of the use of NPT, there is no consensus regarding pressures, dressing change frequency or treatment time, since it is the responsibility of the professional to evaluate these properties based on the characteristics of the wound that is being treated, the tolerability and comorbidities of the patient and the evolution of the lesion. Among the advantages that have been found for NPT, we can highlight its rapid rate of granulation, the preventive and effective approach to infections, the variety and malleability of the dressings, its application at any age and medical history and the possibility of using it with other therapies to accelerate wound closure.

## Data Availability

Not applicable.

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
