# Peer review of "The Effectiveness of Negative Pressure Therapy: Nursing Approach"

_jpm, 2022, doi:10.3390/jpm12111813_

Round 1

Reviewer 1 Report

Thank you for giving me a chance to review it. 

This is a meaningful paper for nursing intervention. This paper is literature review paper, so authors have to describe with logical steps of literature review.

First of all, all sentence of abstracts should be described with full sentence. The first sentence of Introduction has to start with capital. All phrases of objective should be described also with full sentences. In methodology part, Some more information are needed to showing research methods logically. 

In Introduction part, each paragraph is very short. It seems poor writing.  So, some paragraph should be merged with same concepts. Especially, Table 1 and following steps with numbering are not fit the introduction part. Please make other section like previous research review or subtitles.  

This study is a literature review. So, in this methodology, the procedures should be described more detail with logic and systemic steps. So, please show them with subtitle in 'Materials and Methods' part.

After revise and modify it, I will review it again. 

Author Response

Dear reviewer,
We appreciate your recommendations that will improve the quality of the manuscript. Thank you very much for your time and dedication.
Regarding your recommendations:
1) First of all, all sentences in abstracts must be described in complete sentences. The first sentence of the Introduction must begin with a capital letter. All objective phrases must also be described in complete sentences. In the methodology part, some more information is needed to show the research methods in a logical way: revised and modified.
2) In the Introduction part, each paragraph is very short. Seems like bad writing. So some paragraph should be merged with the same concepts. Especially, Table 1 and the following numbered steps do not fit the introductory part. Make another section such as Review of Previous Research or subheadings: Revised and Modified. Subtitles have been created to clarify the argument.
3) This study is a review of the literature. So, in this methodology, the procedures should be described in more detail with logical and systemic steps. So, show them with subtitles in the 'Materials and methods' part: revised and modified. New subtitles have also been created.
Thank you very much for your attention.
We hope that the article is correct for its acceptance.

Reviewer 2 Report

I respectfully recommend that the authors make the following changes:

1), Translate all references from Spanish to English.

2). Use the English acronym for negative pressure therapy (NPT) throughout their manuscript, as opposed to reverting to the Spanish version (TPN-Terapia de Presion Negativa) in the latter half of the report.

3). I recommend a review by a native English speaker, as there are some minor syntax issues that need resolution.

4). This is otherwise an excellent paper and, once the above is corrected, it will be very helpful to the reader.

Author Response

Dear reviewer,
We appreciate your recommendations that will improve the quality of the manuscript. Thank you very much for your time and dedication.
As for your recommendations:
1) Translate all references from Spanish to English: modified and corrected.
two). Use the English acronym for negative pressure therapy (NPT) throughout the manuscript, instead of going back to the Spanish version (TPN-Terapia de Preson Negativa) in the second half of the report: modified and corrected. Also corrected in the abbreviations section.
3). I recommend a review by a native English speaker, as there are some minor syntax issues that need resolution: further review has been sent to the University's language department.
Thank you very much for your attention.
We hope that the article is correct for acceptance.

Round 2

Reviewer 1 Report

According to reviewer's comments, you followed and revised current paper.